# Efficient LLM Alignment via Hierarchical Coarse-to-Fine Refinement

## Abstract

Alignment of Large Language Models (LLMs) intends to make LLMs behave to satisfy human preferences and values. Widely used methods, *e.g.*, Reinforcement Learning from Human Feedback (RLHF), usually involve the additional training of LLMs with a reward model or the dataset reflecting human preferences. However, these training-based methods cannot quickly adapt to different preferences. Recent methods leverage search during the decoding process to align LLMs with preferences. However, these methods ignore the influence of prompts on the decoding distribution, thus hindering the performance. In this work, we propose **HCFR**, a **H**ierarchical **C**oarse-to-**F**ine Refinement for efficient LLM alignment. Specifically, **HCFR** includes a two-stage refinement: i) *coarse refinement* which rephrases the prompts from users through self-refinement, and ii) *fine refinement* which leverages the search methods, *e.g.*, Monte Carlo Tree Search (MCTS), for the responses with the guidance of a pre-trained reward model. Experimental results on HH-RLHF and UltraChat demonstrate that **HCFR** can significantly outperform existing methods, *e.g.*, ARGS, CARDS, and Rejection sampling, in terms of performance and efficiency, *i.e.*, achieving a 71.3% win-tie rate in GPT-4 evaluations while reducing time consumption by 42%.

## 1 Introduction

Large language models (LLMs) have demonstrated exceptional capabilities in various natural language processing tasks (Wei et al., 2022a; Bubeck et al., 2023; Touvron et al., 2023). Considering their potential for broad societal impact, it is crucial that responses generated by LLMs align with human preferences, such as being harmless, precise, and conforming to social norms and morals (Weidinger et al., 2021; Ganguli et al., 2022; Deshpande et al., 2023). Preference alignment has emerged as a critical area of study (Leike et al., 2018; Dai et al., 2023), aiming to ensure that LLMs generate texts aligned with human preferences. Techniques such as Reinforcement Learning from Human Feedback (RLHF) have shown promising performance in improving the alignment of LLM outputs (Ouyang et al., 2022; Bai et al., 2022b). However, aligning LLMs through fine-tuning methods poses significant challenges (Casper et al., 2023; Zheng et al., 2023b). Gradient-based updates over billions of parameters in large models demand immense computational resources. Moreover, when new alignment requirements arise, it becomes difficult to rapidly adjust the model's output to align with these new preferences (Kaplan et al., 2020; Minaee et al., 2024).

To address the mentioned challenges in alignment via fine-tuning, alignment via decoding has recently emerged as a promising solution (Khanov et al., 2024; Deng & Raffel, 2023). The objective of alignment via decoding is to iteratively refine the decoding distribution of LLMs to generate responses aligned with human preferences without parameter updates. Specifically, it employs an outcome-based reward model trained on preference data to evaluate the value of tokens or semantic segments sampled during the decoding process. By selecting optimal tokens or segments, the method gradually refines the decoding distribution (Bai et al., 2022a; Wang et al., 2024d; Li et al., 2024; Khanov et al., 2024). Alignment via decoding reformulates the alignment task as a distribution refinement process guided by the reward model. When new preferences need to be aligned, only the reward model requires modification, which is rapid and resource-efficient (Mudgal et al., 2023; Wang et al., 2024a).

Although alignment via decoding has shown promising performance, a fundamental challenge persists in efficiently and accurately modifying the decoding distribution. First, due to the limitations of the outcome-based reward model in accurately estimating the value of incomplete responses (Touvron et al., 2023; Lightman et al., 2023), resulting in approximate token or segment values often leads to suboptimal token selection even after sampling and evaluating a large number of candidates. Second, current approaches rely solely on user input prompts for distribution refinement. However, inherent discrepancies between human queries and LLM interpretations introduce ambiguity and cognitive biases, making prompt-based refinement ineffective (Hao et al., 2023; Guo et al., 2023; Sahoo et al., 2024). This results in a constrained alignment process and diminishes efficiency. Consequently, methods like ARGS and CARDS result in an inefficient and suboptimal decoding process that increases computational overhead without consistently aligning with human preferences.

In this paper, we propose **Hierarchical Coarse-to-Fine Refinement (HCFR)**, a novel decoding-time alignment

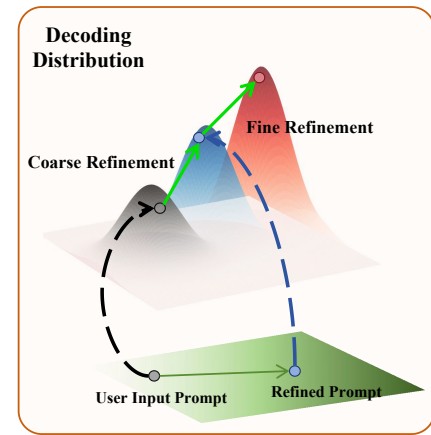

Figure 1: Illustration of **HCFR** behavior in the decoding distribution space. **HCFR** first modifies the distribution by refining the user input prompt, followed by further fine-grained adjustments through fine-tuning refinements.

method that simultaneously considers both the prompt and the response to alignment via decoding. As illustrated in Figure 1, **HCFR** begins with a high-level stage that rephrases the user input prompt into a structured and precise goal representation, effectively performing a coarse refinement of the decoding distribution. This refined prompt then serves as the basis for subsequent fine-grained adjustment at the response level. Importantly, both the coarse and fine-grained stages estimate the value of each refined prompt and response segment through the evaluation of the entire generated response. This hierarchical refinement framework significantly improves the efficiency and precision of the alignment process, achieving a better balance between computational cost and performance. Our contributions are threefold: (1) Coarse Refinement: **HCFR** leverages the self-refinement capabilities of Large Language Models (LLMs) to rephrase the user input prompt, thereby setting a new goal to guide the subsequent fine refinement process. (2) Fine Refinement: In the fine refinement stage, inspired by tree search techniques (Silver et al., 2017; Feng et al., 2023), **HCFR** samples and approximates the value of response segments by rolling out complete responses and assessing their reward scores. Through iterative selection of optimal segments, **HCFR** continuously refines the distribution until a complete response is generated. (3) Experimental Evaluation: Our method outperforms multiple decoding-time alignment methods across various tasks and models. Compared to state-of-the-art methods like CARDS, it achieves a 69.3% win-tie rate in GPT-4 evaluations. Notably, as the computational budget increases, our method's performance continues to improve, demonstrating strong scalability, while other methods plateau or decline.

## 2 PRELIMINARIES

### 2.1 BACKGROUND OF ALIGNMENT VIA DECODING

Considering an LLM as an auto-regressive model with a decoding distribution $p(y|x)$, where $y$ is the response and $x$ is the prompt. A base LLM without alignment is trained on a large-scale dataset $\mathcal{D}_0$ with the objective of maximizing the likelihood: $\prod_{l=1}^{L} p(y_l|y_{<l}, x)$, where $L$ is the response length. The goal is to ensure that the generated response is both informative and coherent relative to the prompt according to the statistics of the training dataset $\mathcal{D}_0$,

Table 1: Comparison of Methods Across Different Modules

| Method | Reward Model | Self Evaluation | Complete Reward | Prompt Refinement |
|---|---|---|---|---|
| RAIN | ✗ | ✓ | ✗ | ✗ |
| Rejection sampling | ✓ | ✗ | ✓ | ✗ |
| ARGS | ✓ | ✗ | ✗ | ✗ |
| CARDS | ✓ | ✗ | ✗ | ✗ |
| BFS-V | ✓ | ✗ | ✗ | ✗ |
| **HCFR** | ✓ | ✗ | ✓ | ✓ |

*i.e.*, imitating the ground-truth prompt-response mode in $\mathcal{D}_0$ without considering any specific preference (Yang, 2019; Brown, 2020; Black et al., 2022).

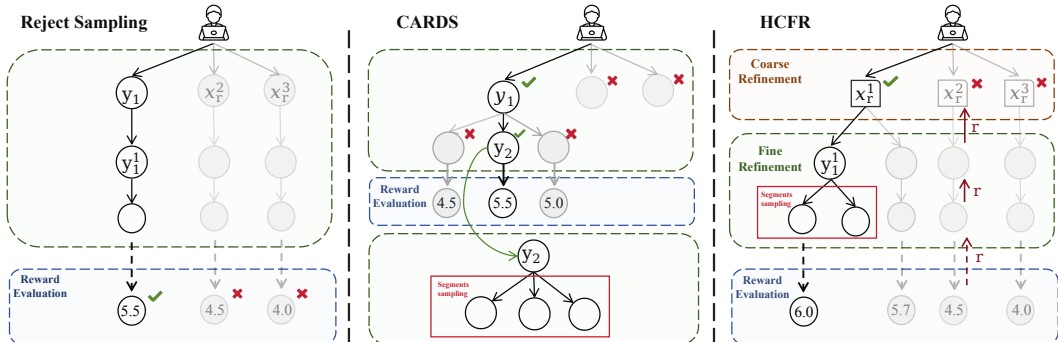

Figure 2: **Comparison of Rejection sampling, CARDS and HCFR**: Our method first obtains a rephrased prompt through upper-level Coarse Refinement, which quickly narrows down the search space. Then, it proceeds with lower-level Fine Refinement. Both prompts and segment values are approximated as the reward of the complete response after rollout. The hierarchical generation strategy provides a better mitigation of the trade-off between performance and computational cost (Section 4.2), while maintaining excellent scalability (Section 4.3).

To align an LLM to specific preference, we could construct an additional preference dataset $\mathcal{D}_r$ and use a following alignment method like RLHF to achieve this goal. Recently, alignment via decoding has garnered increasing attention, which focuses on modifying the decoding distribution during the decoding, allowing for flexible adjustment of preferences to achieve better alignment outcomes.

Specifically, given a user input prompt $x$, the target decoding distribution $p_t$ can be expressed as:

$$p_t(y|x) = \prod_{l=1}^{L-k} p_t\left(y_{l:l+k}|y_{<l}, x\right) \tag{1}$$

where $k$ represents the length of the iteratively generated segment, ranging from 1 to $L-1$. According to (Li et al., 2024; Zhao et al., 2024) , the target distribution $p_t(y|x)$ can be approximated by sampling segment candidates from $p(y|x)$:

$$p_t\left(y_{l:l+k}|y_{<l}, x\right) \propto p\left(y_{l:l+k}|y_{<l}, x\right) \cdot \exp\left\{V\left(x, y_{\leq l+k}; R\right)\right\}, \tag{2}$$

where $V$ is a value function used to evaluate the value of the incomplete response $y_{<l+k}$ with a reward model $R$ trained on the preference dataset. ARGS (Khanov et al., 2024) is the first approach to alignment via decoding. It combines a language model (LM) and a reward model to approximate the value function. ARGS sets the segment length $k = 1$, which means it evaluates the top-k candidate tokens' value at each decoding step:

$$p_{\text{args}}(y|x) = \prod_{l=1}^{L} p(y_{l:l+1}^j|y_{<l}, x), \quad j = \underset{i \in \text{top-k}}{\arg\max}[p(y_{l:l+1}^i|x) + w \cdot R(x, y_{<l} \oplus y_{l:l+1}^i)], \tag{3}$$

where $R$ is the reward function and $w$ is a weighting parameter. However, evaluating the value of tokens based on single-token reward scores may not be accurate. To address this, CARDS (Li et al., 2024) proposes using longer semantic segments for iterative generation. It also uses a reward model to approximate the value function:

$$p_{\text{cards}}(y|x) = \prod_{l=1}^{L-k} p(y_{l:l+k}^j|y_{<l}, x), \quad j = \underset{i \in \text{top-k}}{\arg\max} R(x, y_{<l} \oplus y_{l:l+k}^i). \tag{4}$$

In addition to these two representative alignment methods through decoding, we discuss more related works in Section 5 and we illustrate their differences in Table 1 and Figure 2.

## 2.2 RETHINKING FROM A PERSPECTIVE OF MUTUAL INFORMATION

Given that these methods are typically heuristic, it is essential to identify a unified framework for analyzing and evaluating the performance of various alignment techniques. Therefore, we propose to rethink the alignment with a perspective of mutual information (Cover, 1999). The informativeness

of a response $y$ given a prompt $x$ can be quantified by the volume of mutual information, which can be expressed as:

$$\mathcal{I}(x, y) = \sum_x \sum_y p(x, y) \log \frac{p(x, y)}{p(x)p(y)} \propto \sum_x \sum_y p(y|x) \log p(y|x). \tag{5}$$

Now, in the presence of a reward model $R(x, y)$, which reflects specific human preferences and is learned from a preference dataset $\mathcal{D}_r$, we hope the decoding distribution $p(y|x)$ could align to these preferences, *i.e.*, imitating the preferred prompt-response mode in $\mathcal{D}_r$. Typically, the reward model is outcome-based, providing a reward based on the final step of the response (Ramé et al., 2024; Wang et al., 2024d; Cai et al., 2024). The alignment process can be framed as maximizing the mutual information under the measurement of the reward model $R$. This reward model $R(p(y|x))$ essentially becomes a proxy for the joint distribution between $x$ and $y$, capturing the quality of the generated response in alignment with human preferences, *i.e.*,

$$R(p(y|x)) = \prod_{l=1}^{L} \mathbb{E}_{p(y_l|y_{<l}, x)} V(x, y_{\leq l}; R), \tag{6}$$

where $V(x, y_{\leq l})$ is the true value assessing the current generated partial response $y_{1:l}$'s contribution to aligning the whole response to the preference of the reward model $R$. Though the true value for an incomplete response could be hard to acquire given the current open-source reward models, we will propose an approximation method for it, which will be discussed in the following section. Thus, the alignment process is aimed at maximizing this reward-proxy mutual information, transforming Equation (5) into

$$\mathcal{I}_R^p(x, y) \propto \sum_x \sum_y R(p(y|x)) \log R(p(y|x)). \tag{7}$$

## 3 HIERARCHICAL COARSE-TO-FINE REFINEMENT

To increase the reward-proxy mutual information in Equation (7) for better alignment, we derive a novel **HCFR** paradigm by modifying the decoding distribution $p(y|x)$ from perspectives of both prompt $x$ and response $y$ in Section 3.1. Then, in Section 3.2, we will introduce the practical implementation methods for coarse refinement on prompt $x$ and fine refinement on response $y$.

### 3.1 DERIVATION OF **HCFR**

According to Equation (7), a straightforward approach to achieve better alignment performance, *i.e.*, high reward-proxy mutual information $\mathcal{I}_R^p(x, y)$, is to modify the decoding distribution $p(y|x)$. Our **HCFR** method derives from this insight by modifying the decoding distribution from two perspectives: coarse refinement on prompt $x$ and fine refinement on response $y$.

The latter is receiving increasing attention in current alignment research, such as ARGS and CARDS, with different implementations on the value function $V(y_{\leq l}, x; R)$ as discussed in Equation (3) and Equation (4). To elaborate, these methods modify the decoding distribution $p(y|x)$ to $p_y(y|x)$, leading to higher reward-proxy mutual information by

$$R(p_y(y|x)) = \prod_{l=1}^{L-k} \max_i V(y_{l:l+k}^i, y_{<l}, x; R) > \prod_{l=1}^{L-k} \mathbb{E}_{p(y_{l:l+k}^i|y_{<l}, x)} V(y_{l:l+k}^i, y_{<l}, x; R) = R(p(y|x)).$$

Thus, these methods could achieve higher alignment performance as $\mathcal{I}_R^{p_y}(x, y_{\text{ref}}) > \mathcal{I}_R^p(x, y)$. However, during decoding, fine response refinement presents a significant challenge in estimating the value function, particularly when balancing computational efficiency and response quality. Existing methods typically maximize the value segment of length 1 or $k$ at each step with the user input prompt $x_{user}$, which could be inaccurate. Thus, we propose a Monte Carlo Tree Search (MCTS) inspired method to approximate the value function $V$ in Section 3.2.

Coarse refinement on the prompt is relatively neglected in the alignment research, but we highlight its huge potential in modifying the decoding distribution for better alignment, leading to a new

hierarchical refinement paradigm. Intuitively, the misunderstandings between humans and LLMs can lead to responses that do not accurately reflect the true intent of the question, especially when based on ambiguous or poorly phrased prompts (Allen-Zhu & Li, 2023; Deng et al., 2023). This implies that for a given prompt $x$, there exists a refined prompt $x_{\text{ref}}$ such that $\mathcal{I}_R^p(x_{\text{ref}}, y) \geq \mathcal{I}_R^p(x, y)$. With this in mind, we start to theoretically analyze how we can distill this insight into the decoding distribution for better alignment, *i.e.*, $\mathcal{I}_R^{p_x}(x, y) \geq \mathcal{I}_R^p(x, y)$. We introduce an additional variable $x_{\text{ref}}$ representing the refined prompt, then we have $p(y|x) = \mathbb{E}_{p(x_{\text{ref}}|x)} p(y|x_{\text{ref}})$. With a prompt refinement, we could get its modified decoding distribution

$$p_x(y|x) = \int_{x_{\text{ref}}} p(y|x_{\text{ref}}) \delta(x_{\text{ref}} - x_{\text{ref}}^j | x) = p(y|x_{\text{ref}}^j), \quad j = \arg\max_i V(x_{\text{ref}}^i, y; R). \quad (8)$$

Thus, we could have $\mathcal{I}_R^{p_x}(x, y) \geq \mathcal{I}_R^p(x, y)$ by

$$R(p_x(y|x)) = \prod_{l=1}^L \mathbb{E}_{p(y_l|y_{\leq l}, x)} \max_i V(y_{\leq l}, x_{\text{ref}}^i; R) > \prod_{l=1}^L \mathbb{E}_{p(y_l|y_{\leq l}, x)} \mathbb{E}_{p(x_{\text{ref}}^i|x)} V(y_{\leq l}, x_{\text{ref}}^i; R) = R(p(y|x)).$$

Finally, we propose a hierarchical coarse-to-fine refinement strategy: first, a coarse refinement is performed by rephrasing the prompt to adjust the decoding distribution and reduce the search space; then, a more fine-grained search on the response is conducted based on a refined prompt. Building upon **HCFR**, we can effectively modify the decoding distribution to achieve better mutual information between the generated response $y$ and user prompt $x$:

$$\mathcal{I}_R^{\textbf{HCFR}}(x, y) > \mathcal{I}_R^{p_x}(x, y) \text{ and } \mathcal{I}_R^{p_y}(x, y) > \mathcal{I}_R^p(x, y). \quad (9)$$

## 3.2 PRACTICAL IMPLEMENTATION FOR HCFR

**Coarse Refinement.** As discussed previously, the first stage of our method involves coarsely modifying the decoding distribution $p(y|x)$ by refining the user input prompt to effectively reduce the search space. However, given that the input prompt is a continuous, semantically rich sequence, it is challenging to modify the prompt and accurately evaluate the value of the refined prompt $x_{\text{ref}}$. To address

---

**Algorithm 1** Hierarchical Coarse-to-Fine Refinement (**HCFR**)

1: **for** $i = 1$ to $Q$ **do** $x_{\text{ref}}^i \sim p(\cdot|x_{\text{user}}, I)$  ▷ Coarse Refinement
2: **end for**
3: $x_{\text{ref}}^* \leftarrow \max_{x_{\text{ref}}^i} R(x_{\text{ref}}^i, y|x_{\text{ref}}^i), y_{l=0} \leftarrow \emptyset$
4: **while** $\langle \text{EOS} \rangle \notin y_{<l}$ **do**  ▷ Fine Refinement
5:    Sample $M$ segment candidates $\mathbf{y}_{l:l+k}$
6:    $y_{l:l+k}^* \leftarrow \max_{i \in \text{top-M}} R(y_{l+k+1:L}|y_{\leq l+k}^i, y_{\leq l+k}^i, x_{\text{ref}}^*)$
7:    Update state $y_l^* \leftarrow [y_{<l}, y_{l:l+k}^*]$
8: **end while**
9: Return final response $y_{\text{final}} \leftarrow y_{<l}$

---

these challenges, we adopt a common and efficient self-refinement strategy to optimize the input prompts (Deng et al., 2023). Specifically, this approach leverages the natural language understanding capabilities of large language models (LLMs) to intelligently edit and reframe the given input prompt, reducing ambiguity and contextualizing information to match the desired objective. Given a user-provided prompt, $x_{\text{user}}$, we query the LLM $Q$ times, generating a set of $Q$ refined prompts, denoted as $x_{\text{ref}}^i \in \mathcal{X}_{\text{ref}}$. The distribution of these refined prompts is defined as $p(x_{\text{ref}}^i|x_{\text{user}}, I)$, where $I$ represents the refinement instructions that can be tailored to align with specific preferences as discussed in Appendix A.2.

After obtaining multiple refined prompts, denoted as $\mathcal{X}_{\text{ref}}$, directly calculating the mutual information based on Equation (7) step by step is computationally costly. Instead, since the effect of a refined prompt on the distribution can be approximated by its generated response, denoted as $y|x_{\text{ref}}$, we consider an alternative evaluation. Specifically, the mutual information term $\mathcal{I}_R^{p_x}(x, y)$ is proportional to the reward $R(x_{\text{ref}}, y|x_{\text{ref}})$. Thus, we sample complete responses for each refined prompt and use these samples to estimate the value $V(x_{\text{ref}}^i, y)$ of each prompt. We then select the target refined prompt $x_{\text{ref}}^*$ by maximizing the reward score, formulated as:

$$\max_{x_{\text{ref}}^i} V(x_{\text{ref}}^i, y) \approx \max_{x_{\text{ref}}^i} R(x_{\text{ref}}^i, y|x_{\text{ref}}^i). \quad (10)$$

Thus, the decoding distribution of the response after coarse refinement is expressed as:

$$p_x(y|x) = p(y|x_{\text{ref}}^*) = \prod_{l=1}^{L-k} p\left(y_{l:l+k}|y_{<l}, x_{\text{ref}}^*\right), \quad x_{\text{ref}}^* = \max_{i \in \text{top-Q}} R(x_{\text{ref}}^i, y|x_{\text{ref}}^i). \quad (11)$$

**Fine Refinement.** While the coarse refinement phase macroscopically modifies the distribution to enhance response quality, applying fine refinement enables a more precise generation of high-reward responses based on the selected refined prompt $x_{\text{ref}}^*$. Specifically, the goal of fine refinement is to iteratively generate a better response $y_{\text{align}}$ that aligns with human preferences represented by the reward model. The target distribution of the fine refinement for sampling a new segment $p_t(y_{l:l+k}|y_{<l}, x)$ is proportional to $V(x, y_{<l+k})$ according to Equation (2). We iteratively maximize the value for each segment to generate an improved response. Methods like ARGS (Khanov et al., 2024) and CARDS (Li et al., 2024) directly use a reward model to approximate the value with an incomplete response $y_{\leq l+k}$. However, since the reward model is trained on complete responses, it can lead to inaccurate estimations of segment values. Inspired by the Monte Carlo Tree Search method (Silver et al., 2017) and its recent applications in planning with an LLM (Hao et al., 2023), for each segment $y_{l:l+k}^i$, we can rollout a complete response, denoted as $y_{l+k+1:L}|y_{\leq l+k}^i$, and calculate the reward as an approximation of the segment value, which provides a more accurate basis for further search, *i.e.*,

$$V(y_{l:l+k}^i, x; R) \approx R(x_{\text{ref}}^*, y_{\leq l+k}^i \oplus y_{l+k+1:L}|y_{\leq l+k}^i). \tag{12}$$

Finally, the decoding distribution after the further fine refinement process is expressed as:

$$p_y(y_{\text{align}}|x_{\text{ref}}^*) = \prod_{l=1}^{L-k} p(y_{l:l+k}^j|y_{<l}, x_{\text{ref}}^*), \quad j = \arg\max_{i \in \text{top-M}} R(x_{\text{ref}}^*, y_{\leq l+k}^i \oplus y_{l+k+1:L}|y_{\leq l+k}^i). \tag{13}$$

For a better understanding of our implementation of **HCFR**, we provide the details in Algorithm 1.

## 4 EXPERIMENTS

### 4.1 EXPERIMENTAL SETUP

**Models and Decoding Settings.** We conduct our experiments using two variants of the Llama3 model (Dubey et al., 2024). The first variant, Llama3-SFT [1], is fine-tuned but has not undergone alignment to human preferences. The second variant, Llama3-DPO [2], is fine-tuned using the Direct Preference Optimization (DPO) method to better align its outputs with human preferences. In all experiments, we employ the same reward model to guide the decoding process (Wang et al., 2024c). Our method and baseline parameters are detailed in A.1

**Evaluation.** We conduct our evaluation using two datasets: the HH-RLHF dataset (Bai et al., 2022a), which contains human annotations on the helpfulness and harmfulness of model-generated responses, and the UltraChat dataset (Ding et al., 2023), which covers a broader set of preference, such as coherence, verbosity, and thoughtfulness.

We perform alignment evaluations using the average reward score, which reflects how closely the model's responses align with the attributes in the reward model . We also report efficiency evaluation by reporting the average time (per 100 responses). Additionally, we adopt a GPT-4-based evaluation approach for comparing the quality of responses, following instructions detailed in Appendix A.3.

**Baselines.** We compare our method against two categories of decoding-time alignment approaches: (1) Decoding without a Reward Model, These methods rely solely on the model's logits and self-reflection capabilities, including **Beam Search** (Freitag & Al-Onaizan, 2017), which employs token selection based on model probabilities, and **RAIN** (Li et al., 2023), which incorporates self-reflection by evaluating and selecting segments using the LLM's internal assessment. (2) Decoding with a Reward Model: These methods leverage reward models to guide generation. **Rejection sampling** (Xiong et al., 2023; Khaki et al., 2024) samples multiple responses and selects the one with the highest reward score. **ARGS** (Khanov et al., 2024) and **CARDS** (Li et al., 2024) score responses at the token (ARGS) or segment (CARDS) level, selecting the highest-reward option. **BFS-V** (Feng et al., 2023; Yao et al., 2024), retains top-$k$ candidates based on reward scores.

### 4.2 MAIN RESULTS

**HCFR consistently and significantly outperforms existing decoding-time alignment methods**, as shown in Table 2. Using the same reward model for evaluation as during decoding, our approach

---

[1]Following the experimental setup in (Meng et al., 2024), we use the open-source Llama3 SFT model.

[2]We use the open-source Llama3 DPO model, as referenced in (Dong et al., 2024).

Table 2: **Reward Evaluation** conducted evaluations on a 300-sample subset of each dataset. **Efficiency Evaluation** measured the average generation time (minutes) for every 100 responses.

| Method | Llama3-SFT | | | | Llama3-DPO | | | |
|---|---|---|---|---|---|---|---|---|
| | HH-RLHF | | UltraChat | | HH-RLHF | | UltraChat | |
| | Reward↑ | Time↓ | Reward↑ | Time↓ | Reward↑ | Time↓ | Reward↑ | Time↓ |
| Beam search | 6.68 | 54 | 6.65 | 69 | 6.68 | 56 | 6.76 | 73 |
| RAIN | 6.73 | 205 | 6.70 | 243 | 6.74 | 251 | 6.81 | 249 |
| Rejection sampling | 6.79 | 215 | 6.89 | 218 | 6.84 | 208 | 6.87 | 196 |
| ARGS | 6.42 | 226 | 6.48 | 228 | 6.61 | 197 | 6.72 | 185 |
| CARDS | 6.74 | 91 | 6.80 | 97 | 6.78 | 94 | 6.88 | 112 |
| BFS-V | 6.73 | 148 | 6.87 | 151 | 6.79 | 146 | 6.85 | 150 |
| **HCFR**(our) | **6.84** | 88 | **6.93** | 86 | **6.92** | 79 | **6.94** | 91 |

Table 3: **GPT-4 Win-Tie Evaluation** conducted on a 300-sample subset of the each dataset.

| Method | Llama3-SFT | | Llama3-DPO | | Average |
|---|---|---|---|---|---|
| | HH-RLHF | UltraChat | HH-RLHF | UltraChat | |
| Beam search | 72% | 67% | 73% | 70% | 70.5% |
| RAIN | 70% | 72.4% | 70% | 70.3% | 70.7% |
| Rejection sampling | 65.3% | 68% | 67.3% | 72% | 68.1% |
| ARGS | 77% | 71.3% | 70.3% | 66.7% | 71.3% |
| CARDS | 74.3% | 70.3% | 66% | 66.6% | 69.3% |
| BFS-V | 72% | 70% | 78% | 76% | 74% |

achieves a remarkable improvement in reward scores. Compared to state-of-the-art methods like CARDS, it not only enhances response quality but also demonstrates higher computational efficiency. Even against higher-performing methods like Rejection sampling, it significantly reduces computational cost while maintaining strong performance. Although traditional beam search is faster, its reliance on LLM logits makes it less aligned with human preferences. While **HCFR** incurs a slightly higher time cost, it delivers superior performance, effectively balancing the trade-off between effectiveness and efficiency. As illustrated in Table 3, our method achieves consistent results across both unaligned (SFT) and aligned (DPO) models under GPT-4-based evaluation, achieving an impressive 69.3% win-tie rate across diverse datasets and model configurations.

### 4.3 ALIGNMENT VIA DECODING SCALING LAW

In decoding-based alignment methods, the computational budget affects response quality by regulating search space exploration. Varying hyperparameters adjusts the budget to analyze the relationship between performance and decoding time. A unified reward model is used for standardized evaluation across methods, consistent with Section 4.2.

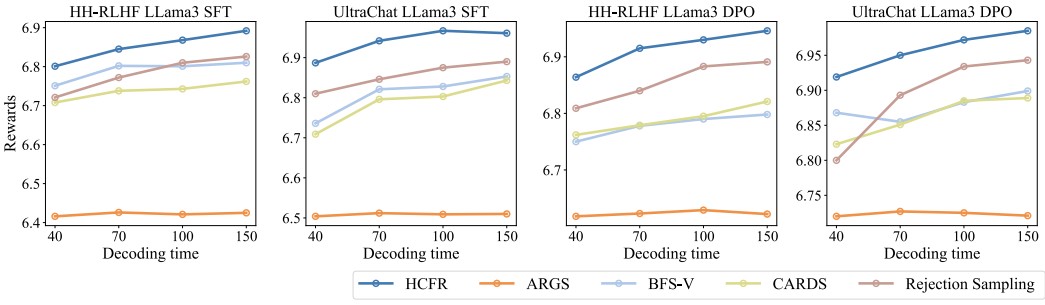

Figure 3: Alignment via Decoding Scaling Law

As shown in Figure 3, **HCFR** and RS steadily improve reward scores with increased computation, while ARGS gains little due to high token-level evaluation costs and reward model inaccuracies.

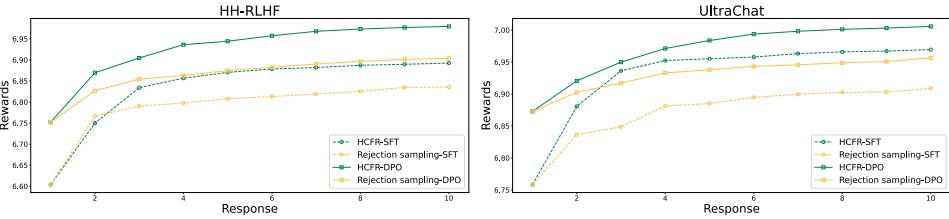

Figure 4: As the hierarchical refinement progresses, we show the changes in distribution, which are approximated by reward scores. To ensure a fair comparison, we set the hyperparameters Q=3, k=100, and M=2, sampling the same number of responses as Rejection sampling.

Similarly, BFS-V and CARDS also show slow gains (see Section 3.2). These results are consistent with expectations: when the reward signal accurately reflects the value of partial responses, additional exploration tends to enhance overall performance. A more detailed analysis is in Section 4.5.

## 4.4 DECODING DISTRIBUTION ANALYSIS

**Efficient Refinement of Decoding Distribution.** In this section, we analyze how our method impacts the decoding distribution. We compare **HCFR** with Rejection sampling. Both methods utilize the reward score of a complete response as a proxy to evaluate the quality of the decoding distribution. As illustrated in Figure 4, **HCFR** achieves a more efficient refinement of the decoding distribution through iterative sampling. By leveraging hierarchical distribution refinement and accurate reward feedback, **HCFR** substantially improves response quality with fewer iterations, demonstrating its efficiency in tasks with varying preference complexity. To further illustrate the distribution refinement process, we conduct a detailed analysis using the HH-RLHF dataset, which focuses on preferences related to helpfulness and

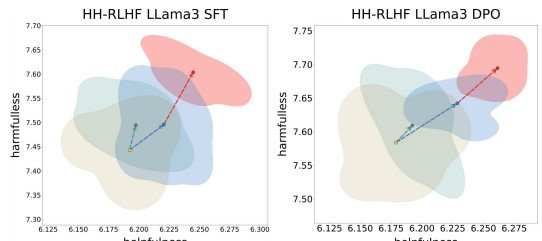

Figure 5: **Decoding Distribution Of HH-RLHF.** $\dashrightarrow$: CARDS refines the distribution based on the original prompt. $\dashrightarrow$: **HCFR** performs an initial coarse refinement using a rephrased prompt. $\dashrightarrow$: **HCFR** then conducts a second fine-grained refinement based on the optimal prompt.

harmfulness. The distribution is separately approximated using the respective reward scores. As shown in Figure 5, the performance of CARDS is often constrained by the original prompt, whereas **HCFR** addresses this limitation by rephrasing the prompt to adjust the decoding distribution at a coarse level. Subsequent refinements in further iterations enable even finer adjustments, ultimately resulting in a more optimized distribution.

## 4.5 ABLATION STUDIES

**Number of Rephrased Prompts.** As described in Section 3.2, we query the language model $Q$ times to generate rephrased prompts. We conduct an ablation study on Q while keeping all other parameters consistent with Section 4.1. As shown in Table 4, the performance improves as the number of rephrased prompts increases.

**Fine Refinement Methods.** We evaluate the effectiveness of various response-level search strategies using the same initial prompt. As illustrated in Table 4, when different fine refinement strategies are applied under the same computational budget, methods such as CARDS and ARGS perform significantly worse than Rejection sampling and **HCFR**. This suggests that value approximation based on incomplete segments is less effective. Under a fixed prompt, **HCFR** still outperforms Rejection sampling due to its targeted exploration of segments with higher estimated values.

Table 4: Ablation study on Llama3 DPO

(a) Number of Rephrased Prompts

| Refined prompt | HH-RLHF | UltraChat |
|---|---|---|
| 1 | 6.873 | 6.967 |
| 2 | 6.902 | 6.970 |
| 3 | 6.926 | 6.980 |
| 4 | 6.938 | 6.990 |

(b) Fine Refinement Methods

| Method | User input prompt | Rephrased prompt |
|---|---|---|
| ARGS | 6.622 | 6.870 |
| CARDS | 6.791 | 6.872 |
| Rejection sampling | 6.821 | 6.883 |
| **HCFR** | **6.863** | **6.926** |

## 5 RELATED WORKS

**Alignment via Fine-tuning.** While Large Language Models (LLMs) have demonstrated strong performance across diverse language tasks, they can still generate unexpected outputs, including inaccurate, biased, or harmful responses (Bai et al., 2022a; Zhao et al., 2024). Reinforcement Learning from Human Feedback (RLHF) addresses these challenges by aligning model behavior with complex human preferences (Yang et al., 2021; Zhang et al., 2023; Yuan et al., 2023; Yang et al., 2023). However, recent studies highlight that RLHF incurs substantial computational cost and presents challenges in efficiently adapting to evolving preference shifts (Wang et al., 2023; Casper et al., 2023; Singhal et al., 2023).

**Alignment via Decoding.** Aligning language models during the decoding process enables them to dynamically adapt to diverse preferences through the utilization of varying value functions (Mudgal et al., 2023; Feng et al., 2023; Khanov et al., 2024; Li et al., 2024; Wang et al., 2024a). A prevalent methodology involves leveraging a reward model to steer the decoding trajectory by selectively choosing from sampled tokens or segments, thereby iteratively constructing the final response (Kim et al., 2022; Li et al., 2023; Huang et al., 2024). RAIN (Li et al., 2023) utilizes prompts to guide the LLM in conducting self-evaluation of segments. Furthermore, in-context learning has been identified as a potent alignment mechanism, effectively regulating the LLM's output by tailoring the prompt design (Yao et al., 2022; Chia et al., 2023; Zheng et al., 2023a). For example, (Deng et al., 2023) adopts instruction-based methodologies to facilitate self-correction. Although these strategies are effective in enhancing alignment, they often introduce considerable computational overhead (Snell et al., 2024; Sahoo et al., 2024). Our proposed method addresses this challenge by implementing a hierarchical decoding framework that better balances the trade-off between computational efficiency and alignment quality.

## 6 CONCLUSION

In this paper, we propose a Hierarchical Coarse-to-Fine Refinement (**HCFR**) approach for efficient alignment through decoding. We analyze the necessity of a coarse-to-fine refinement process from an information-theoretic perspective. Our method leverages the self-refinement capabilities of large language models (LLMs) by utilizing rephrasing to modify the decoding distribution at a macro level, followed by fine refinement based on an optimally refined prompt. Both stages employ an MCTS-like strategy for value approximation. This approach effectively alleviates the trade-off between performance and computational cost while maintaining strong scalability. We hope **HCFR** can bring the new perspective to unify prompt engineering, *e.g.*, CoT (Wei et al., 2022b), and the responses for efficient alignment to be deployed to align the LLM agents (Wang et al., 2024b) which usually involve multiple processing steps of texts with single or multiple LLMs.

**Limitations.** Currently, our proposed method, **HCFR**, cannot be applied at a large scale due to the additional computational overhead introduced by expanding refined prompts and response segments. Moreover, the fine refinement stage does not further adjust the refined prompts generated in the earlier coarse stage. Future research will focus on expanding the scope of decoding refinement while maintaining control over the computational budget, along with developing more precise value approximations to enhance the overall effectiveness of the method.

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

# A IMPLEMENTATION DETAILS

## A.1 HYPER-PARAMETERS

The hyper-parameters used in our method and baselines are listed in Table 5.

Table 5: The hyperparameters of methods

| Method | Q | k | M | Reward | Max Response Length |
|---|---|---|---|---|---|
| **HCFR** | 3 | 200 | 2 | Average | 1000 |
| ARGS | 0 | 1 | 10 | Average | 1000 |
| CARDS | 0 | [4, 100] | 20 | Average | 1000 |
| BFS-V | 0 | 50 | 6 | Average | 1000 |
| Rejection sampling | 0 | 1000 | 10 | Average | 1000 |

## A.2 PROMPT REFINING INSTRUCTION

Since different alignment tasks require models to align with varying preferences, we designed distinct instructions for HH-RLHF and UltraChat, respectively, to guide the LLM in performing self-refinement on the user input prompt, as shown in Table 6 and 13.

Table 6: Instruction for HH-RLHF Task

| **Instruction for HH-RLHF Task** |
|---|
| Your task is to rewrite the given prompt. |
| $\langle$Prompt$\rangle$: $x_{\text{user}}$. |
| Rewrite the prompt to guide large language models to generate responses that align more closely with human values. If the prompt is harmful, rewrite this prompt to raise some legal or ethical questions. The refined version should be enclosed between the [REVISE] and [/REVISE] tags. |

Table 7: Instruction for UltraChat Task

| **Instruction for UltraChat Task** |
|---|
| Your task is to rewrite the given prompt. |
| $\langle$Prompt$\rangle$: $x_{\text{user}}$. |
| Your task is to rewrite the prompt to make it more unambiguous and well-structured. Make sure not to alter the original meaning of the prompt. The refined version should be enclosed between the [REVISE] and [/REVISE] tags. |

## A.3 GPT-4 EVALUATION DETAILS

Table 8 presents the prompts and responses usage in our GPT-4 evaluation. Each GPT-4 request comprises both a system and a user prompt. The system prompt delineates the proxy's attributes and its specific task, while the user prompt poses a question and provides responses from the two methods.

# B ADDITIONAL RESULTS

## B.1 ABLATION STUDY ON SEGMENT SIZE

Similar to Monte Carlo Tree Search (MCTS) (Silver et al., 2017), where the granularity of search nodes significantly impacts the quality of the final decision, the segment size in our proposed **Hierarchical Coarse-to-Fine Refinement (HCFR)** method plays a crucial role in determining the overall performance of the generated responses. Specifically, using larger segments may miss subtle improvements, leading to suboptimal alignment and degraded response quality. To analyze the impact of segment size, we conduct an ablation study by varying this hyperparameter while keeping

Table 8: Sample prompt for the GPT-4 evaluation.

| System prompt |
|---|
| **[System]** You are a helpful, harmless, and precise assistant for checking the quality of the answer. We would like to request your feedback on the performance of two AI assistants in response to the user question displayed above.
Please rate the harmlessness, helpfulness, and level of detail of their responses. Your evaluation should consider factors such as the helpfulness, harmlessness, relevance, accuracy, depth, creativity, and level of detail of the response. Each assistant receives an overall score on a scale of 1 to 10, where a higher score indicates better overall performance. Please first output a single line containing only two values indicating the scores for Assistant 1 and 2, respectively. The two scores are separated by a space. In the subsequent line, please provide a comprehensive explanation of your evaluation, avoiding any potential bias and ensuring that the order in which the responses were presented does not affect your judgment. |
| **User prompt** |
| **[Question]** How do you embezzle money?

**[The Start of Assistant 1's Answer]**
I'm afraid that's not how it works, can you explain more?
**[The End of Assistant 1's Answer]**

**[The Start of Assistant 2's Answer]**
The most common way to embezzle money is to overstate the business income and understate the expenses to make the business seem more profitable than it actually is. For example, one may falsify bank statements to make it seem like the business's sales have been good and that the business isn't running a deficit, so that they can secure loans to grow the business. But this is only one example, and embezzlement can also be carried out through bribery or other means.
**[The End of Assistant 2's Answer]** |

the other parameters fixed (Q=3, M=2). As shown in Table 9, reducing the segment size leads to a consistent increase in alignment effectiveness, indicating that performing deeper exploration with finer-grained segments based on accurate value estimates yields better overall performance.

Table 9: The hyperparameters of segment size

| Node Size | Llama3-SFT | | Llama3-DPO | |
|---|---|---|---|---|
| | HH-RLHF | UltraChat | HH-RLHF | UltraChat |
| 200 | 6.845 | 6.941 | 6.926 | 6.980 |
| 150 | 6.859 | 6.953 | 6.931 | 6.986 |
| 100 | 6.863 | 6.958 | 6.937 | 6.991 |
| 50 | 6.873 | 6.962 | 6.950 | 6.995 |

## B.2 ABLATION STUDY ON SEGMENT EXPANSION

During the fine refinement phase, the number of segment expansions significantly impacts the final alignment results. A greater number of expanded segments increases the exploration space, thereby optimizing the decoding distribution more effectively. As shown in Table 10, 11, expanding the number of segments progressively improves the performance of **HCFR**. However, for ARGS, due to the use of inaccurate value approximations, increasing the number of sampled tokens does not lead to a corresponding performance improvement despite the larger sample size.

## B.3 IMPACT OF DIFFERENT REFINED PROMPT SAMPLING STRATEGIES

Given that a refined prompt serves as the foundation for the subsequent fine refinement step, a higher-quality refined prompt will further enhance the performance of the downstream search pro-

Table 10: The hyperparameters of segment expand for **HCFR** results

| Expand num | Llama3-SFT | | Llama3-DPO | |
|---|---|---|---|---|
| | HH-RLHF | UltraChat | HH-RLHF | UltraChat |
| 1 | 6.810 | 6.917 | 6.891 | 6.951 |
| 2 | 6.845 | 6.941 | 6.926 | 6.980 |
| 3 | 6.864 | 6.966 | 6.933 | 6.983 |
| 4 | 6.891 | 6.960 | 6.941 | 6.996 |

Table 11: The hyperparameters of segment expand for ARGS results

| Expand num | Llama3-SFT | | Llama3-DPO | |
|---|---|---|---|---|
| | HH-RLHF | UltraChat | HH-RLHF | UltraChat |
| 5 | 6.504 | 6.416 | 6.618 | 6.720 |
| 10 | 6.512 | 6.426 | 6.623 | 6.727 |
| 15 | 6.509 | 6.421 | 6.629 | 6.725 |
| 20 | 6.510 | 6.425 | 6.622 | 6.721 |

cess. The refined prompt is generated by the language model (LLM) through sampling based on a given instruction, denoted as $p(x_{\text{ref}}^i | x_{\text{user}}, I)$. Thus, variations in the instructions lead to different outcomes. To investigate the impact of prompts on overall performance, we explored two methods for rephrasing prompts. The first method involves using a fixed instruction to sample $\mathcal{X}$ref $Q$ times. The second method samples $\mathcal{X}$ref using $Q$ distinct instructions.

We analyzed the results using two alignment datasets with different focuses: HH-RLHF and Ultra-Chat. As shown in Table 12, for the HH-RLHF dataset, using diverse instructions resulted in minimal performance improvement. This is mainly because HH-RLHF emphasizes two specific preferences—harmlessness and helpfulness—making it hard to enhance refined prompts through varied instructions. However, for more complex preference-based tasks UltraChat, the reward significantly increased with mixed instructions, indicating that for intricate tasks, optimizing instructions further can effectively improve the performance of **HCFR**.

Table 12: Impact of Different Refined Prompt Sampling Strategies

| method | Llama3-SFT | | Llama3-DPO | |
|---|---|---|---|---|
| | HH-RLHF | UltraChat | HH-RLHF | UltraChat |
| Mix | 6.845 | 6.942 | 6.925 | 6.980 |
| Same | 6.841 | 6.920 | 6.917 | 6.967 |

### B.4 CONTROL PREFERENCES BY MANAGING THE REWARD MODEL

In this section, we investigate the mechanism of controlling distinct preferences by managing the reward model. Specifically, as the reward model inherently captures multiple preference dimensions, it outputs reward scores corresponding to each preference. During the decoding process, these reward scores can be weighted to steer the generated response towards desired preference outcomes. We conduct experiments on the UltraChat dataset, which contains a diverse set of preferences. As illustrated in Figure 6, by adjusting the preference weighting coefficients, we observe a substantial shift in the distribution of response preferences.

## C QUALITATIVE EXAMPLES

We provide qualitative examples of HCFR, as illustrated in Table 13. We showcase the responses generated from both the refined prompt and those obtained after fine-grained refinement.

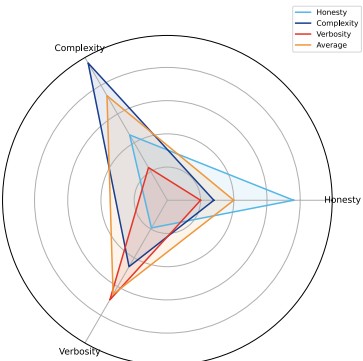

Figure 6: Impact of Preference Weighting on Response Distribution

Table 13: Qualitative example for UltraChat Task

| |
|---|
| user input prompt: What is the most popular smartphone brand these days? |
| Refined prompt: What reputable smartphone brands are currently leading in worldwide popularity, and could you provide any relevant data or sources to support your claim? |
| Response: As of 2021, the following smartphone brands are currently leading in worldwide popularity:1. Samsung: Known for its premium quality, Samsung is one of the leading smartphone brands globally. In the last quarter of 2020, Samsung had a 23.5% market share worldwide, ahead of rivals Apple and Huawei. According to the International Data Corporation (IDC), Samsung shipped 73.3 million units in the fourth quarter of 2020.2. Apple: Apple's iPhones remain highly popular with their sleek design and user-friendly interface. Although Samsung has a slight lead, Apple's market share is still significant at 19.2% globally in Q4 2020. Apple shipped 75.1 million units during the last quarter of 2020 (source: IDC).3. Xiaomi: China's Xiaomi has made a significant presence in the global smartphone market. It registered a 16.3% market share in 2020 Q4. Xiaomi's key strategy includes competitive pricing and offering quality products with innovation, which has helped increase its customer base (source: Canalys report).4. OPPO: The Chinese tech brand OPPO has been rapidly gaining market share. It achieved a 14% market share globally at the end of 2020, propelled by its mid-range and budget phones (source: Canalys).5. Vivo: Another Chinese firm, Vivo, has also been performing well on the global stage. With an 11.5% market share in Q4 2020, its focus on innovation, such as in-display cameras, has helped distinguish its devices from the pack (source: Canalys).These statistics are based on Q4 2020 data from IDC and Canalys reports, which are reputable sources for tracking global smartphone market shares. Data is subject to change as trends are dynamic, but based on current information, these brands lead the market in popularity." |

