# OpenReview forum: "Efficient LLM Alignment via Hierarchical Coarse-to-Fine Refinement"
_ICLR.cc/2025/Conference — ICLR 2025 Conference Withdrawn Submission_

### Official Review · Reviewer_3JHQ · 2024-10-27

**Soundness:** 2
**Presentation:** 3
**Contribution:** 3
**Rating:** 5
**Confidence:** 4

**Summary:**

This paper works on decoding time alignment.
The proposed method is a two-stage approach. In the first stage, the responses undergo a rephrasing according to user-specified prompt.
In the second stage, decoding time search is used.
Experiments conducted on two tasks are used to demonstrate the effectiveness of the proposed method, comparisons are made against other decoding-time alignment baselines.

**Strengths:**

The presentation of the paper is good. The illustrative figures, algorithm, related tables are helpful in guiding readers.
Experiment visualizations are clear.

**Weaknesses:**

The necessity of the two-stage method is not highlighed enough. Hence those stages are somewhat isolated to each other.

The method is not compared with fine-tuning / prompting-based methods. Although this work focuses on decoding-time alignment, it would be useful for readers to understand the pros and cons of the proposed method by having comparisons to alternative approaches. To be specific, what would the training effort differences (e.g., the reward model and value model training in decoding and the reward models in fine tuning approaches), what are the computational resources needed? Noted that I'm not expecting one method to be better than the other all the time, but a more comprehensive understanding of multiple aspects of different methos would be helpful.

The idea and format of having Table 1 is great, but the current version is not clear enough why those properties are necessary. Further explaination could enhance the clarity.

**Questions:**

Can the authors explain why the two-stage method is needed? And what is the benefit of the proposed method in merging those stages as compared to existing approaches?

Why do the authors report win-and-tie as the evaluation metric? This metric can be very misleading if the tie-rate is higher than half of the win-tie-rates.

Learning accurate reward/value function is essential for effective MCTS, could the authors explain more on their efforts about how robust is their method to different reward models? (ideally, one will need precise token-level reward models such that the search can be correctly guided). In practice, how challenge is it to generate such reward model or value estimators in MCTS?

What is the exploration strategy in the MCTS implementation? How do the authors banlance between limited searching budget and the performance? Could the authors show the scalability ablation study on the MCTS step? Ideally, when the number of explorative nodes increases, we could see the performance gain ("inference time scaling law" as a fancy name) across different searching budgets.

---

### Official Review · Reviewer_5N1b · 2024-11-02

**Soundness:** 2
**Presentation:** 2
**Contribution:** 2
**Rating:** 3
**Confidence:** 4

**Summary:**

The authors introduce a Hierarchical Coarse-to-Fine Refinement (HCFR), a decoding-time alignment method for Large Language Models (LLMs) that integrates both the prompt and response into a structured, efficient alignment process. HCFR operates in two stages.

Coarse Refinement: this rephrases the user's input prompt into a structured goal, refining the decoding distribution at a high level.

Fine Refinement that leverages the MCTS method to refine the complete response segments further while assessing their reward scores and iteratively selecting optimal segments until a full response is generated.

The authors claim that this hierarchical framework improves alignment accuracy and efficiency by balancing computational cost with performance.

**Strengths:**

The problem addressed in this paper is both exciting and novel. The analysis appears sound, and the proofs seem correct at first glance. However, I have some questions I would like to clarify, as highlighted below.

**Weaknesses:**

Though the paper addresses a good problem, it still lacks some details, and I would like to see more clarity in the revised versions.

Terms like 'efficient', 'balanced,' and 'refined' are somewhat vague and could be more specific. For instance, mentioning how HCFR improves efficiency quantitatively or qualitatively would provide a clearer picture of its impact.

The contributions seem marginal, modifying the decoding distribution $p(y|x)$ for both prompt $ x $ and response $ y $, given that this process is well-studied for the response.

**Questions:**

1. In terms of practical implementation, HCFR still struggles to handle large-scale problems. I would like to see a comparison with methods where prompt optimization is performed separately, and refinement occurs solely at the response level.

2. In line 222, you reference $x_{\text{ref}}$​ and mention that the expectation is taken over these reference prompts. However, it’s unclear how this reference prompt, $x_{\text{ref}}$​, is obtained.

3. It seems that HCFR uses the instruction prompt I to generate $x_{\text{ref}}$​, which resembles the concept of a "meta prompt" commonly discussed in prompt optimization literature. Could you clarify why prompt-level HCFR is necessary in this context?

4. Equation 5 is presented at a very high level. I would ike to see the proportionality constants used here.

---

### Official Review · Reviewer_LdUF · 2024-11-04

**Soundness:** 2
**Presentation:** 4
**Contribution:** 3
**Rating:** 5
**Confidence:** 4

**Summary:**

- This paper introduces HCFR, a hierarchical method for improving decoding time alignment with human preferences.
 - HCRF works in 2 stages:
      - Coarse refinement: Refines the prompt to be more precise and includes necessary context.
      - Fine refinement: MCTS during decoding to generate a more optimal final response as measured by the reward model.

 - Coarse and fine alignments are decoupled i.e. coarse alignment is done first, and the aligned prompts are used for the finer alignment.
 - HCRF is much more efficient and required 42% less time compared to other baselines and qualitatively improves upon the SOTA decoding time alignment methods.

**Strengths:**

- The approach is novel and explores a novel approach for decoding time alignment -- prompt refinement.
 - It is more efficient and results in responses that are more aligned as measured by the chosen RM.
 - The method's performance is customizable -- it improves with increased computational budget and changes as the RM weights are modified.
 -  Both coarse and fine refinement seem to be complementary, yet distinct in their role.

**Weaknesses:**

- In general, experimental details are sparse. For example, there is very little information about the reward model used and it is central to the effectiveness of this method. It is unclear how the choice of the RM affects the effectiveness of this method.
 -  The authors also don't mention the size of the Llama 3 model used and it’s not clear how this method would work for larger or smaller models.
 - It's unclear at what point increasing the number of prompts for coarse refinement yields diminishing returns, and whether potential gains from the combined coarse and fine refinement are limited by either stage.

**Questions:**

Mentioned in the weaknesses.

---

### Note · Authors · 2024-12-02

I have read and agree with the venue's withdrawal policy on behalf of myself and my co-authors.